# N-Cadherin mRNA Levels in Peripheral Blood Could Be a Potential Indicator of New Metastases in Breast Cancer: A Pilot Study

**DOI:** 10.3390/ijms21020511

**Published:** 2020-01-14

**Authors:** Takaaki Masuda, Hiroki Ueo, Yuichiro Kai, Miwa Noda, Qingjiang Hu, Kuniaki Sato, Atsushi Fujii, Naoki Hayashi, Yusuke Tsuruda, Hajime Otsu, Yosuke Kuroda, Hidetoshi Eguchi, Shinji Ohno, Koshi Mimori, Hiroaki Ueo

**Affiliations:** 1Department of Surgery, Kyushu University Beppu Hospital, 4546 Tsurumihara, Beppu 874-0838, Japannmiwa-1126@oita-u.ac.jp (M.N.); guitar5158@gmail.com (Q.H.); basement.kuni13@gmail.com (K.S.); afujii2003@yahoo.co.jp (A.F.); isaya_hikoan@yahoo.co.jp (N.H.); yxmms497@ybb.ne.jp (Y.T.); ootsu@surg2.med.kyushu-u.ac.jp (H.O.); kuro731976@yahoo.co.jp (Y.K.); heguchi@beppu.kyushu-u.ac.jp (H.E.); 2Department of Surgery, Saiseikai Karatsu General Hospital, 817 Motohata, Karatshu 847-0852, Japan; ueohiro@med.kyushu-u.ac.jp; 3Ueo Breast Surgical Hospital, 188-2 haya, Oita 870-0854, Japan; kai@oita-mamma.jp; 4Breast Oncology Center, The Cancer Institute Hospital Ariake of Japanese Foundation for Cancer Research, 3-8-31, Ariake, Koto, Tokyo 135-8550, Japan; shinji.ohno@jfcr.or.jp

**Keywords:** N-cadherin, EMT, breast cancer, new metastasis, eribulin, blood, biomarker

## Abstract

Background: There is growing evidence that patients with metastatic breast cancer whose disease progresses from a new metastasis (NM) have a worse prognosis than that of patients whose disease progresses from a pre-existing metastasis. The aim of this pilot study is to identify a blood biomarker predicting NM in breast cancer. Methods: The expression of epithelial (cytokeratin 18/19) or mesenchymal (plastin-3, vimentin, and N-cadherin) markers in the peripheral blood (PB) of recurrent breast cancer patients undergoing chemotherapy with eribulin or S-1 was measured over the course of treatment by RT-qPCR. The clinical significance of preoperative N-cadherin expression in the PB or tumor tissues of breast cancer patients undergoing curative surgery was assessed by RT-qPCR or using public datasets. Finally, N-cadherin expression in specific PB cell types was assessed by RT-qPCR. Results: The expression levels of the mesenchymal markers N-cadherin and vimentin were high in the NM cases, whereas that of the epithelial marker cytokeratin 18 was high in the pre-existing metastasis cases. High preoperative N-cadherin expression in PB or tumor tissues was significantly associated with poor recurrence-free survival. N-cadherin was expressed mainly in polymorphonuclear leukocytes in PB. Conclusion: N-cadherin mRNA levels in blood may serve as a novel prognostic biomarker predicting NM, including recurrence, in breast cancer patients.

## 1. Introduction

Distant metastasis is the leading cause of mortality in patients with cancer, including breast cancer, which is the most common malignancy in women worldwide despite the advent of new treatments [1]. Thus, controlling distant metastasis is important for prolonging survival.

Recently, the importance of the type of distant metastasis progression, namely new metastasis (NM) versus growth of a pre-existing metastasis (PEM), has been highlighted, because the progression type can affect the prognosis of patients with metastatic cancer [2,3,4] (Appendix A). Interestingly, patients with recurrent breast cancer who develop NM have a worse prognosis than that of those with PEM [2,3,5,6]. However, the differences between these two progression types has not affected treatment determination, because both NM and PEM are classified clinically as a “progressive disease” (PD) according to the diagnostic criteria of the Response Evaluation Criteria in Solid Tumors [7]. There have also been cases of NM vs. PEM reported in colorectal and lung cancers [2] and liposarcoma [4]. This concept of distant metastasis progression stems from phase 3 clinical trials of eribulin treatment of metastatic breast cancer [8,9].

Eribulin mesylate (eribulin; Eisai Co., Tokyo, Japan) is a non-taxane microtubule inhibitor with a novel mechanism of action involving irreversible blockade of mitosis at the G2/M phase, followed by apoptosis [10]. Eribulin is currently approved for the treatment of certain patients with advanced breast cancer in many countries worldwide, including Japan [8]. Interestingly, two different phase 3 clinical trials of patients with metastatic breast cancer showed that eribulin has more pronounced effects on overall survival compared with progression-free survival [8,9]; one possible explanation is that eribulin suppresses the incidence of NM, thus providing an increased survival benefit to patients. The preclinical studies described herein were designed to assess whether eribulin has such an anti-metastatic property, via reversion of epithelial–mesenchymal transition (EMT) [11,12]. Yoshida et al. and Terashima et al. showed experimentally that eribulin suppresses metastasis of breast cancer cells by inducing conversion of EMT to mesenchymal–epithelial transition (MET) [11,13]. EMT is the process by which epithelial cells lose their cell-cell junctions and acquisition of front-rear polarization, resulting in the formation of mesenchymal cells with migratory properties, and it is characterized by loss of the epithelial markers E-cadherin and cytokeratins (CKs), together with increased expression of mesenchymal markers such as N-cadherin and vimentin. EMT is observed during cancer progression; it promotes invasion and metastasis by facilitating the motility of tumor cells [14,15]. These data suggest that EMT may induce NM.

Thus, we hypothesized that monitoring the EMT status in real-time can predict NM and is important for guiding the treatment of patients with recurrent breast cancer. We focused on liquid biopsy, which is used to identify biomarkers in body fluids, mainly blood, and provides non-invasive real-time information regarding tumor characteristics [16,17]. In this study, we monitored the expression of EMT (mesenchymal) and MET (epithelial) markers in the peripheral blood (PB) of patients with recurrent breast cancer undergoing chemotherapy with eribulin or S-1 (oral 5-fluorouracil derivative) and identified N-cadherin as a useful marker predicting NM. Furthermore, we assessed the clinical significance of preoperative N-cadherin expression in the PB of breast cancer patients undergoing curative surgery.

## 2. Results

### 2.1. Type of Metastasis Progression in Patients with Recurrent Breast Cancer who underwent Chemotherapy with Eribulin or S-1

The subjects comprised 56 and 19 patients who underwent chemotherapy with eribulin and S-1, respectively. Of the 56 patients treated with eribulin, 35 received S-1 prior to eribulin. The patient characteristics are shown in Table 1.

As shown in Table 2, the patients treated with eribulin had a significantly lower incidence of NM than patients treated with S-1, although the time to treatment failure (TTF) was shorter in the former ((a) in Table 2, *p* = 0.043). Moreover, the patients treated with S-1 followed by eribulin also had a significantly lower incidence of NM under eribulin than under S-1 ((b) in Table 2, *p* = 0.025). These results support previous clinical and experimental findings that eribulin suppresses NM via conversion of EMT to MET in tumor cells [11,12,13]. Furthermore, these findings led us to hypothesize that markers of EMT may be predictive of NM, which we tested using samples from the patients treated with eribulin or S-1.

### 2.2. Expression of Epithelial and Mesenchymal Markers in Breast Cancer and Non-Epithelial Cell Lines

First, we examined if our selected markers reflect the EMT status by evaluating their expression in various cell lines. The quantitated gene expression levels of the different markers are shown in Figure 1a. The epithelial markers (CK18 and CK19) were expressed in all breast cancer cell lines, whereas the mesenchymal markers (PLS3, vimentin, and N-cadherin) were expressed mainly in the non-epithelial cell lines. PLS3, which is reportedly expressed in both epithelial and mesenchymal cells [18], was expressed in most of the cell lines albeit at varying levels among them.

### 2.3. Expression of Epithelial and Mesenchymal Markers in Breast Cancer Tissues

We also assessed the expression levels of the epithelial and mesenchymal markers in breast cancer tissues using The Center Genome Atlas (TCGA) datasets (Figure 1b,c). As shown in Figure 1b, CK18, CK19, and N-cadherin levels were higher in tumor tissues (*n* = 1093) than in normal tissues (*n* = 112) of breast cancer patients (*p* < 0.001). Unexpectedly, PLS3 and vimentin expression levels were lower in tumor tissues than in normal tissues (*p* < 0.001).

Next, we compared marker expression between ER-positive (*n* = 823) and ER-negative (*n* = 219) cases (Figure 1c). As expected, the expression levels of the mesenchymal markers were higher in the ER-negative than ER-positive cases (PLS3, vimentin, and N-cadherin: *p* < 0.001, *p* < 0.001, and *p* = 0.002, respectively), whereas the expression levels of the epithelial markers were higher in the ER-positive than ER-negative cases (CK18 and CK19: both *p* < 0.001). These results suggest that mesenchymal markers are expressed in high-grade cancers with metastatic potential, because ER-negative tumors tend to be associated with earlier relapse and worse prognosis compared with ER-positive tumors [19,20,21,22].

### 2.4. Expression of Epithelial and Mesenchymal Markers in the PB of Breast Cancer Patients

Next, we assessed the mRNA expression levels of the epithelial and mesenchymal markers in the PB of 16 patients with recurrent breast cancer and 10 healthy volunteers (HVs) using Ueo and Beppu cohorts (Figure 2a). CK18, vimentin, and N-cadherin expression in PB was statistically higher in the patients with recurrent breast cancer than in HVs (*p* = 0.031, *p* = 0.004, and *p* = 0.031, respectively). Other markers also had a tendency to be higher in PB from patients compared with HVs. These findings indicate that these markers are expressed in circulating tumor cells (CTCs) or host cells in the PB of breast cancer patients.

### 2.5. Expression of Epithelial and Mesenchymal Markers in the PB of Patients with Recurrent Breast Cancer with NM or PEM

We compared the expression levels of epithelial and mesenchymal markers in the PB of patients with recurrent breast cancer with NM (the total number of samples; *n* = 9 from 4 patients) versus PEM (the total number of samples; *n* = 16 from 3 patients) using Ueo cohort. The changes in the expression levels of the markers in a representative case of NM or PEM are shown in Figure 2b. Interestingly, the expression level of N-cadherin increased consistently with time in the NM case, although the level of the tumor markers CEA and CA15-3 were not elevated. In the PEM case, the expression of N-cadherin was low, although CA15-3 was elevated. Furthermore, the expression levels of the mesenchymal markers tended to be higher in the NM, whereas epithelial marker expression was higher in the PEM cases (Figure 2c). Statistical differences in N-cadherin, vimentin, and CK18 levels between the NM and PEM cases were found (*p* = 0.002, *p* = 0.027, and *p* = 0.011, respectively). These data suggest that N-cadherin expression in PB is predictive of NM, and N-cadherin may reflect the real-time metastatic potential of tumor cells.

### 2.6. Expression of Epithelial and Mesenchymal Markers in the PB of Patients with Recurrent Breast Cancer undergoing Eribulin or S-1 Rreatment

Next, we compared the levels of epithelial and mesenchymal markers in the PB of patients with recurrent breast cancer undergoing eribulin (the total number of samples; *n* = 19 from 4 patients) versus S-1 (the total number of samples; *n* = 17 from 8 patients) treatment (Figure 2d). Interestingly, the expression of N-cadherin in PB was statistically lower in the patients treated with eribulin than in those treated with S-1 (*p* < 0.001).

Moreover, the expression level of N-cadherin was decreased by eribulin treatment compared with before treatment. The changes in N-cadherin expression in the breast cancer patients over the course of eribulin or S-1 treatment are shown in Figure 2e. Among the eight patients treated with S-1, six showed over a twofold increase in the N-cadherin expression compared with baseline, whereas only one of the four patients treated with eribulin showed over a twofold increase in N-cadherin expression compared with baseline. These data further support that eribulin can induce the conversion of EMT to MET, as reported previously [11,12,13].

### 2.7. Clinicopathological Significance of Preoperative N-Cadherin mRNA Expression in the PB of Breast Cancer Patients undergoing Curative Surgery

The results of the abovementioned expression analyses motivated us to investigate the value of preoperative N-cadherin expression in PB for predicting breast cancer recurrence in patients after curative surgery, because recurrence is considered a type of NM.

First, we assessed N-cadherin expression in the PB of the 326 patients with breast cancer using Kyushu cohort. The N-cadherin levels ranged from 5.805 × 10^−5^ to 0.352 (median, 0.045). The median expression level of N-cadherin in patients with TNM stage I, II, or III (126, 182, or 16 cases, respectively) was 0.042, 0.048, or 0.051, respectively (Figure 3a). The median expression levels of N-cadherin in the ER-positive (*n* = 240) and ER-negative (*n* = 83) cases were 0.045 and 0.047, respectively. There was no significant difference in the N-cadherin level in PB according to TNM stage or ER status (Figure 3a,b).

Next, the relationships between clinicopathological factors and N-cadherin expression in blood were examined in the 326 patients with breast cancer. The patients were divided into two groups (high and low N-cadherin expression) as described in Materials and Methods. The cutoff level of N-cadherin expression was 0.046. As shown in Table 3, there were no significant differences between the high and low N-cadherin expression groups in terms of clinicopathological factors including age, tumor size, nuclear grade, venous involvement, lymphatic involvement, lymph node metastasis, ER/PgR/HER2 status, and subtype.

### 2.8. Prognostic Significance of Preoperative N-Cadherin mRNA Expression in the PB of Breast Cancer Patients undergoing Curative Surgery

Next, we assessed the prognostic significance, in terms of RFS, of N-cadherin expression in PB using Kyushu cohort. The high N-cadherin expression group (*n* = 159) had a significantly worse recurrence-free survival (RFS) (*p* = 0.041) than the low N-cadherin expression group (*n* = 167) (Figure 3c). Next, univariate and multivariate regression analyses of predictive factors for RFS were performed (Table 4). Univariate analyses showed that lymphatic involvement, lymph node metastasis, PgR-negative, and high N-cadherin expression were statistically significant prognostic factors for RFS (*p* = 0.002, *p* = 0.002, *p* = 0.026, and *p* = 0.040, respectively). N-cadherin expression and clinicopathological factors such as lymph node metastasis, and PgR status were included in the multivariate analysis. N-cadherin expression in PB was not a significant independent prognostic factor for RFS in patients with breast cancer according to the multivariate analysis (HR: 2.215, *p* = 0.092). 

### 2.9. Prognostic Significance of N-Cadherin mRNA Expression in the Tumor Tissues of Breast Cancer Patients undergoing Curative Surgery

We investigated the prognostic significance of N-cadherin in the tumor tissues of breast cancer patients treated with curative surgery using the Kaplan–Meier plotter dataset. “Systemically untreated breast cancer patients” with any intrinsic subtype were selected for the analysis. These patients were divided into two groups based on N-cadherin expression by selecting the “Auto Select best cutoff” feature of the Kaplan–Meier plotter. As expected, the RFS was shorter in the high (*n* = 1173) than low N-cadherin expression group (*n* = 2778) among all cases (*p* < 0.001, 1% FDR, Figure 3d).

These clinical results suggest that N-cadherin expression in PB reflects the metastatic potential of tumor cells and is a potential biomarker predicting NM in breast cancer (Figure 3e).

### 2.10. Assessment of N-Cadherin mRNA Expression in PB Cell Fractions

Finally, we assessed the distribution of N-cadherin expression in PB cells of breast cancer patients using Beppu cohort. The expression levels of N-cadherin and CKs in polymorphonuclear leukocytes (PMNs), mononuclear cells (MCs), and red blood cells (RBCs) from 24 patients with breast cancer were measured by RT-qPCR (Figure 4). The average ratios of N-cadherin relative expression in RBCs, MCs and PMNs to that in whole blood were 0.021, 0.417, and 14.375, respectively. N-cadherin expression was much higher in PMNs than in MCs or RBCs (*p* < 0.001), indicating that N-cadherin is expressed predominantly in PMNs within PB (Figure 4a).

Next, we compared N-cadherin expression in PMNs and MCs between breast cancer patients (n = 24) and HVs (*n* = 10) (Figure 4b). In PMNs, the median levels of N-cadherin, CK18, and CK19 expression were 1.698, 0.706, and 0.350 in breast cancer patients and 0.851, 0.249, and 0.167 in HVs, respectively. In MCs, the median levels of N-cadherin, CK18, and CK19 expression were 0.264, 0.079, and 3.214 in breast cancer patients and 0.195, 0.062, and 2.228 in HVs, respectively. The expression levels of N-cadherin, CK18, and CK19 in PMNs were higher in breast cancer patients than in HVs (*p* = 0.004, *p* = 0.003, and *p* = 0.003, respectively); however, in MCs, there was no statistical difference in expression levels between breast cancer patients and HVs. These findings suggest the possibility that N-cadherin-expressed cells are circulating tumor cells (CTCs) in PMN fraction of breast cancer patients because CKs-expressed cells in PB are mainly cancer cells in patients with various epithelial malignancies [23].

## 3. Discussion

N-cadherin is a transmembrane glycoprotein that mediates calcium-dependent cell-cell adhesion, mediated by post-translational modifications such as phosphorylation of the N-cadherin catenin complex [24]. This protein is multifunctional and can interact with many different proteins and be involved in many different functional events [25]. N-cadherin possesses seven glycation sites on its ectodomain with ectodomains. Therefore, in order to obtain a real picture about its function it is necessary to consider that: i) its adhesive function can be certainly regulated by gradual post-translational modifications; ii) for example, its structural organization can strongly depend on its phosphorylation status; and iii) its N-glycosylation sites can regulate N-cadherin-dependent cell adhesion. These points are very interesting and need further investigations to elucidate which form of the N-cadherin complex in blood can play a role in NM development.

In this pilot study, we demonstrated that N-cadherin expression in PB is a potentially useful biomarker for predicting NM, which is associated with a poor prognosis in breast cancer, by sequential monitoring of N-cadherin expression levels in the PB of patients undergoing chemotherapy. To our knowledge, this is the first study to identify a biomarker predictive of NM. Furthermore, we observed that breast cancer patients treated with eribulin, compared with S-1, have a low incidence of NM and low expression of the mesenchymal marker N-cadherin in PB. This adds to the growing evidence that eribulin suppresses NM by conversion of EMT to MET in tumor cells. 

Cadherins are single-chain transmembrane glycoproteins that mediate calcium-dependent homophilic cell-cell adhesion and play a critical role in regulating signaling pathways that maintain essential gene transcription [26]. N-cadherin, one of most well-studied cadherins, is expressed mainly in neural and mesenchymal tissues. In various malignancies including breast cancer, cells acquire motility and invasiveness by upregulating N-cadherin during EMT [27,28,29]. We also reported that N-cadherin is associated with tumor aggressiveness in esophageal carcinoma [30], and others demonstrated the value of N-cadherin as a marker of invasive, malignant tumors [29,31], supporting our finding that N-cadherin expression in PB may be a predictive biomarker of NM. The expression level of N-cadherin may represent the real-time metastatic potential of tumor cells (Figure 3e). 

Our prognostic analysis of N-cadherin expression in PB and tumor tissues showed that high preoperative levels are associated with early recurrence in breast cancer patients undergoing curative surgery. These observations provide new insight that recurrence in primary breast cancer after curative surgery is a type of NM. Furthermore, this insight suggests that sequential monitoring of N-cadherin expression in PB during postoperative follow-up may help determine the best treatment for patients with breast cancer.

We also evaluated the specific cell type in PB that expresses the highest level of N-cadherin in breast cancer patients. PMNs expressed the highest level of N-cadherin compared with MCs and RBCs. Furthermore, expression of N-cadherin as well as the epithelial markers CK18 and CK19 in PMNs was higher in breast cancer patients than in HVs, implying that N-cadherin is expressed mainly in CTCs, similar to vimentin [32]. However, PMNs may express N-cadherin upon stimulation by tumor cells. Further study is required to clarify the biological significance of N-cadherin expression in PMNs and identify other PB cell types expressing N-cadherin.

There were some limitations to this study. First, a larger cohort is needed to validate the clinical utility of sequential monitoring of N-cadherin expression in PB. Next, monitoring N-cadherin mRNA levels in PB may not be suitable in the clinic because of the complication of the measurement system. Measurement of N-cadherin protein levels may be more practical because proteins in blood are easily measured by ELISA, as shown in a previous study that evaluated N-cadherin levels in the blood of patients with malignant melanoma [33]. Finally, we focused only on PLS3, vimentin, and N-cadherin, among the various EMT markers, as potential biomarkers of NM. Based on this pilot study, we plan to perform comprehensive analyses in PB using RNA sequencing or mass spectrometry.

## 4. Materials and Methods

### 4.1. Study Patients

The data from recurrent breast cancer patients treated with eribulin (56 cases) and/or S-1 (19 cases) at Ueo Breast Surgical Hospital (Oita, Japan) from January 2015 to December 2017 were extracted from the medical records. We chose the data from patients treated with S-1 (oral 5-fluorouracil derivative) as a reference control to compare with those treated with eribulin. This is because both S-1 and eribulin are often used as third-line chemotherapy agents, after anthracyclines and taxanes [1], for patients with recurrent breast cancer in Japan, based on the findings of a randomized controlled trial (SELECT BC) showing equivalent overall survival between S-1-treated and taxane-treated patients with metastatic or recurrent breast cancer [34]. Furthermore, in contrast to eribulin, 5-fluorouracil was reported to induce the conversion of MET to EMT [11,13]. Clinical data, including the progression of metastases during either eribulin or S-1 treatment, from breast cancer patients were also investigated. In this study, disease progression was defined as NM and/or growth of PEM based on systemic computed tomography findings.

### 4.2. Patient Cohorts in Gene Expression Analysis

We used 3 patient cohorts (Ueo, Beppu, and Kyushu) and 2 public datasets (The Cancer Genome Atlas (TCGA) and The Kaplan–Meier Plotter) for expression analysis. The details are described below. All patients provided written informed consent to participate in this study, which was approved by each institutional review board and the Ethics and Indications Committee of Kyushu University (#577-00, #29-597).

### 4.3. Clinical Samples

To observe genes expression levels over time in blood, PB samples were processed from blood, collected during routine examinations conducted over a 1-year period (2015–2016), from 20 patients with HER2-negative breast cancer with relapse undergoing chemotherapy (eribulin or S-1) at Ueo Breast Surgical Hospital (affiliated with Kyushu University Beppu Hospital), after obtaining written informed consent (Ueo cohort). Inclusion criteria were Eastern Cooperative Oncology Group grade ≤ 2 and adequate organ and hematological function. Eribulin was administered intravenously (1.4 mg/m^2^) on days 1 and 8 of each 3-week cycle. S-1 was administered orally (40–60 mg) twice daily for 28 consecutive days, followed by a 14-day rest period.

Observation was continued until the patient experienced disease progression, according to systemic computed tomography findings, or after 1 year of chemotherapy with eribulin or S-1 if no disease progression was detected. The detailed clinical characteristics and sampling protocol are provided in Appendix A and Appendix A. Immediately after collection, each 1 mL sample of blood was mixed with 4 mL ISOGEN-LS (Nippon Gene, Toyama, Japan) and stored at −80 °C until RNA extraction. Sixteen of the blood samples from 16 cases were sent to Kyushu University Beppu Hospital for analysis without knowledge of the histopathological or clinical results. Four cases dropped out in the middle of this study. Samples from 12 cases total were processed for gene expression analysis.

Furthermore, gene expression was evaluated in three different fractions of blood cells: polymorphonuclear leukocytes (PMNs), mononuclear cells (MCs), and red blood cells (RBCs) (Beppu cohort). Blood samples were obtained from 24 breast cancer patients with invasive carcinoma who underwent primary tumor resection at Kyushu University Beppu Hospital in 2017. Control blood samples were obtained from 10 healthy volunteers (HVs) at Kyushu University Beppu Hospital. The blood samples were immediately processed for isolation of the three cell types.

To determine the clinical significance of preoperative gene expression in the PB of patients with primary breast cancer undergoing curative surgery, we used clinical data from breast cancer patients described previously (Kyushu cohort) [35]. Briefly, a total of 594 patients with breast cancer underwent primary tumor resection at the Department of Breast Oncology, National Kyushu Cancer Center (Fukuoka, Japan), from 2000 to 2008. Of these, 356 female patients with breast cancer without distant metastases, preoperative therapy, or previous treatment for other cancers were included in this study. Among these patients, 326 with invasive carcinoma were included in a survival analysis. The observation period ranged from 0.3 to 6.9 years (median 3.8 years). Postoperative adjuvant therapy was performed according to the guidelines set by the St. Gallen Consensus Conference [36]. The patients underwent clinical examinations at least every 3 months and mammography annually and were further evaluated only if they exhibited symptoms.

The stages and grades of the tumors were classified according to the AJCC/UICC TNM classification and stage groupings. All data including age, pathological tumor size, nuclear grade, venous involvement, lymphatic involvement, lymph node metastasis, and the statuses of estrogen receptor (ER), progesterone receptor (PgR), and human epidermal growth factor receptor (HER2) expression were obtained from medical records. Recurrence-free survival (RFS) was defined as the period from surgical treatment for cancer to detection of any sign of recurrence.

### 4.4. Cell Lines

Twelve human cell lines were used in this study: CRL1500, MCF-7, MDA-MB231, Mrknu1, SKBR3, YMB1 (all breast intraductal carcinoma cell lines), HMEC (mammary epithelial cell line), Raji (B lymphocytes), Jurkat (T lymphocytes), HT1080 (fibrosarcoma), THP-1 (monocytes), and KMST-6 (fibroblasts). The latter five are all non-epithelial tumor cell lines. Among the breast cancer cell lines, CRL1500 and MCF7 cells are ER positive, MDA-MB231 cells are ER, PgR and HER2 negative (triple negative), Mrknu1 cells are ER negative, SKBR3 cells are ER negative and HER2 positive, and the ER/PgR/HER2 status of YMB1 cells is unknown. The cell lines were obtained from the Cell Resource Center for Biomedical Research Institute of Development, Aging and Cancer, Tohoku University, and were maintained in RPMI-1640 supplemented with 10% fetal bovine serum at 37 °C in a 5% humidified CO_2_ atmosphere. Upon reaching a subconfluent state, the cell cultures were homogenized and the lysates stored at −80 °C.

### 4.5. Separation of Three Blood Cell Fractions

We used Polymorphprep^TM^ (Alere Technologies AS, Oslo, Norway) to isolate three blood cell fractions (PMNs, MCs, and RBCs) according to the manufacturer’s instructions, as described previously [35]. Each fraction was mixed with ISOGEN II (Nippon Gene) and stored at −80 °C until RNA extraction.

### 4.6. Total RNA Extraction

Total RNA was extracted from cell lines or blood samples using ISOGEN II or ISOGEN-LS, respectively, according to the manufacturer’s instructions (Nippon Gene, Toyama, Japan).

### 4.7. Mesenchymal and Epithelial Markers

Plastin-3 (PLS3), vimentin, and N-cadherin were selected for evaluation as mesenchymal markers [37]. PLS3 is a marker of circulating tumor cells, including not only epithelial cells but also tumor cells undergoing EMT [18,38]. CK18 and CK19 were selected for evaluation as epithelial markers, as the controls [23,37].

### 4.8. RT-qPCR

RT-qPCR of CK18, CK19, PLS3, vimentin, N-cadherin, RNA18S5, and GAPDH mRNA levels was performed as described previously [39]. In brief, reverse transcription was performed using random hexamers and M-MLV reverse transcriptase (Invitrogen, Carlsbad, CA, USA). qPCR was performed using LightCycler^®^ FastStart DNA Master SYBR Green I (Roche Diagnostics, Basel, Switzerland). The raw data are presented as the relative cDNA level from Human Universal Reference Total RNA (Clontech Laboratories, Palo Alto, CA, USA) and normalized to the level of the internal control gene (RNA18S5 or GAPDH). Relative quantification of gene expression was calculated using the 2^−ΔΔCt^ method. The primer sequences used for RT-PCR were as follows: CK18, forward 5′-ATCTTGGTGATGCCTTGGAC-3′ and reverse 5′-CCTGCTTCTGCTGGCTTAAT-3′; CK19, forward 5′-CATGAAAGCTCCCTTGGAAGA-3′ and reverse 5′-TGATTCTGCCGCTCACTATCAG-3′; PLS3, forward 5′-CCTTCCGTAACTGGATGAACTC-3′ and reverse 5′-GGATGCTTCCCTAATTCAACAG-3′; vimentin, forward 5′-TACAGGAAGCTGCTGGAAGG-3′ and reverse 5′-ACCAGAGGGAGTGAATCCAG-3′; N-cadherin, forward 5′-ATTGGACCATCACTCGGCTTA-3′ and reverse 5′-CACACTGGCAAACCTTCACG-3′; RNA18S5, forward 5′-AGTCCCTGCCCTTTGTACACA-3′ and reverse 5′-CGATCCGAGGGCCTCACTA-3′; and GAPDH, forward 5′-TTGGTATCGTGGAAGGACTC-3′ and reverse 5′-AGTAGAGGCAGGGATGATGT-3′.

### 4.9. TCGA Analysis

We used TCGA data to analyze the expression of CK18, CK19, PLS3, vimentin, and N-cadherin in breast cancer tissues. The expression data of these genes in tumor tissues from 1093 breast cancer cases and in normal tissues from 112 cases with breast cancer available in TCGA were obtained from the Broad Institute’s Firehose pipeline (http://gdac.broadinstitute.org/runs/stddata__2016_01_28/data/BRCA/20160128/). The sequencing data were normalized by quantile normalization, as described previously [40].

### 4.10. Kaplan–Meier Plotter Survival Analysis

The Kaplan–Meier Plotter (www.kmplot.com), an online database that includes gene expression and clinical datasets, was used to generate Kaplan–Meier plots for RFS as described previously [41].

### 4.11. Statistical Analysis

For the clinical analysis, the cases were divided into two groups according to N-cadherin expression using the minimum *p* value approach, which is a comprehensive method to determine the optimal cutoff point for survival risk classification among continuous gene expression measurements from multiple datasets [42]. The variables were compared using the Mann–Whitney U test or Fisher’s exact test. Survival curves were generated using the Kaplan–Meier method and compared by log-rank test. Cox proportional hazards regression was used for univariate and multivariate analyses to calculate hazard ratios for the factors associated with survival. A two-sided *p* < 0.05 was deemed statistically significant. Statistical analyses were performed using JMP Pro 13 software (SAS Institute, Cary, NC, USA).

## 5. Conclusions

We demonstrated that N-cadherin mRNA expression in blood serves as a novel prognostic biomarker for predicting NM and cancer recurrence in patients with breast cancer. Furthermore, we provide clinical evidence that eribulin inhibits NM possibly via suppression of EMT. These findings highlight the role of N-cadherin in NM, which serves as a guide for the treatment of breast cancer, and provide a better understanding of the molecular mechanism of breast cancer recurrence.

## Figures and Tables

**Figure 1 ijms-21-00511-f001:**
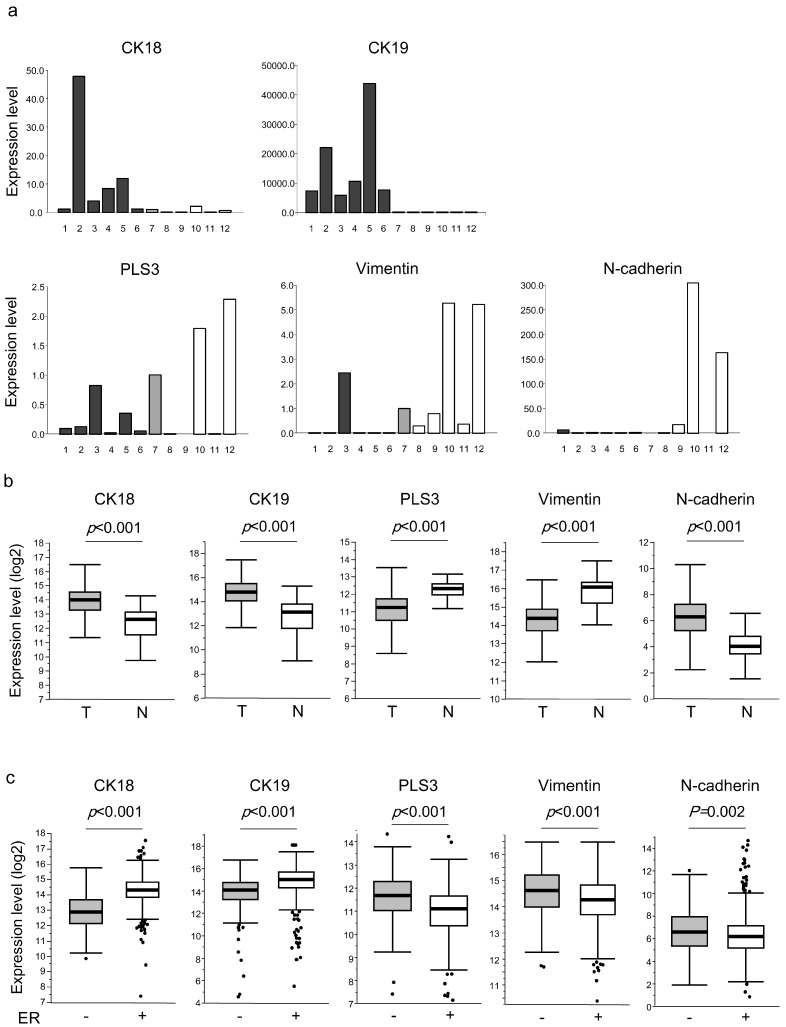
Expression of epithelial and mesenchymal markers in breast cancer cell lines and tissues. (**a**) Expression in invasive ductal carcinoma cell lines (#1–6: CRL1500, MCF-7, MDA-MB231, Mrknu1, SKBR3, and YMB1 cells, respectively), a normal mammary epithelial cell line (#7: HMECs), and non-epithelial cell lines (#8–12: Raji B lymphocytes, Jurkat T lymphocytes, HT1080 fibrosarcoma cells, THP-1 monocytes, KMST-6 fibroblasts, respectively). The expression levels are expressed relative to the level in HMECs (1.0). (**b**) Expression in normal and tumor tissues of breast cancer patients obtained from The Center Genome Atlas (TCGA) dataset. T: tumor tissues; N: normal mammary tissues. (**c**) Expression in tissues of breast cancer patients from TCGA dataset according to estrogen receptor (ER) status. +: ER-positive; −: ER-negative.

**Figure 2 ijms-21-00511-f002:**
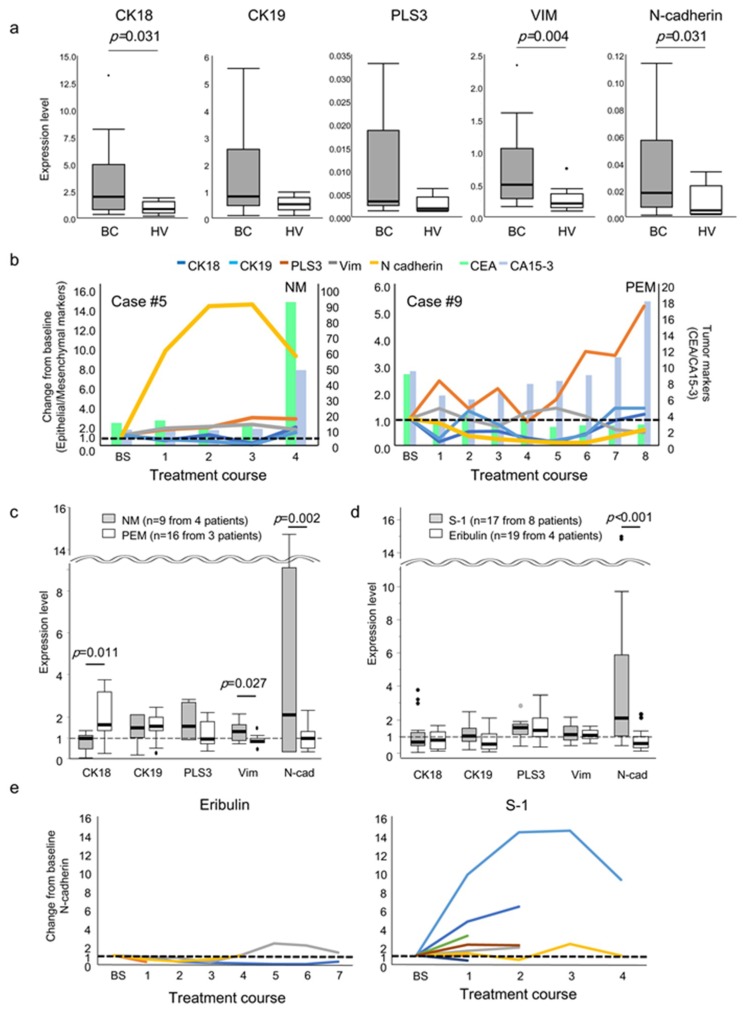
Expression of epithelial and mesenchymal markers in the peripheral blood (PB) of breast cancer patients undergoing chemotherapy. (**a**) A comparison of expression levels of the markers in PB between breast cancer patients just before eribulin or S-1 chemotherapy and healthy volunteers (HVs). (**b**) Changes in the expression levels of the markers in representative cases of NM and PEM over the treatment course of eribulin or S-1 treatment. Left: case #5 treated with S-1; right: case #9 treated with eribulin. The expression levels are expressed relative to the level at pre-treatment (baseline; 1.0). BS; baseline. (**c**) A comparison of expression levels of the markers in PB of breast cancer patients with NM or PEM. The expression levels are expressed relative to the level at pre-treatment (baseline; 1.0). BS; baseline. n; the total number of samples from patients. (**d**) A comparison of expression levels of the markers in PB of breast cancer patients undergoing eribulin or S-1 treatment. The expression levels are expressed relative to the level at pre-treatment (baseline; 1.0). BS; baseline. n; the total number of samples from patients. (**e**) Changes in N-cadherin expression in breast cancer patients over the course of eribulin or S-1 treatment. The expression levels are expressed relative to the level at pre-treatment (baseline; 1.0). BS; baseline.

**Figure 3 ijms-21-00511-f003:**
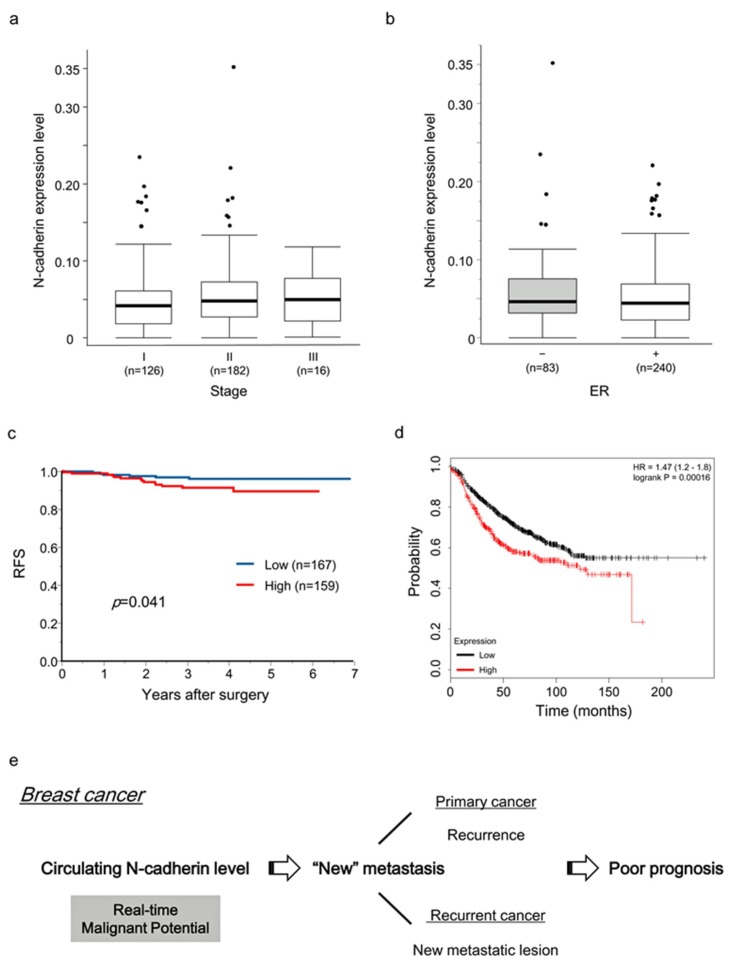
Preoperative expression levels of N-cadherin mRNA in the PB of breast cancer patients undergoing curative surgery. (**a**) N-cadherin expression in the PB of patients with breast cancer according to TNM stage. (**b**) N-cadherin expression in the PB of patients with breast cancer according to ER status. +: ER-positive; −: ER-negative. (**c**) The RFS of 326 patients with breast cancer after curative surgery according to N-cadherin expression in preoperative PB. (**d**) The RFS of 3951 patients with invasive ductal carcinoma from the Kaplan–Meier plotter dataset according to N-cadherin expression in breast cancer tissues. (**e**) A proposed model showing the clinical significance of the circulating N-cadherin level for predicting NM in breast cancer.

**Figure 4 ijms-21-00511-f004:**
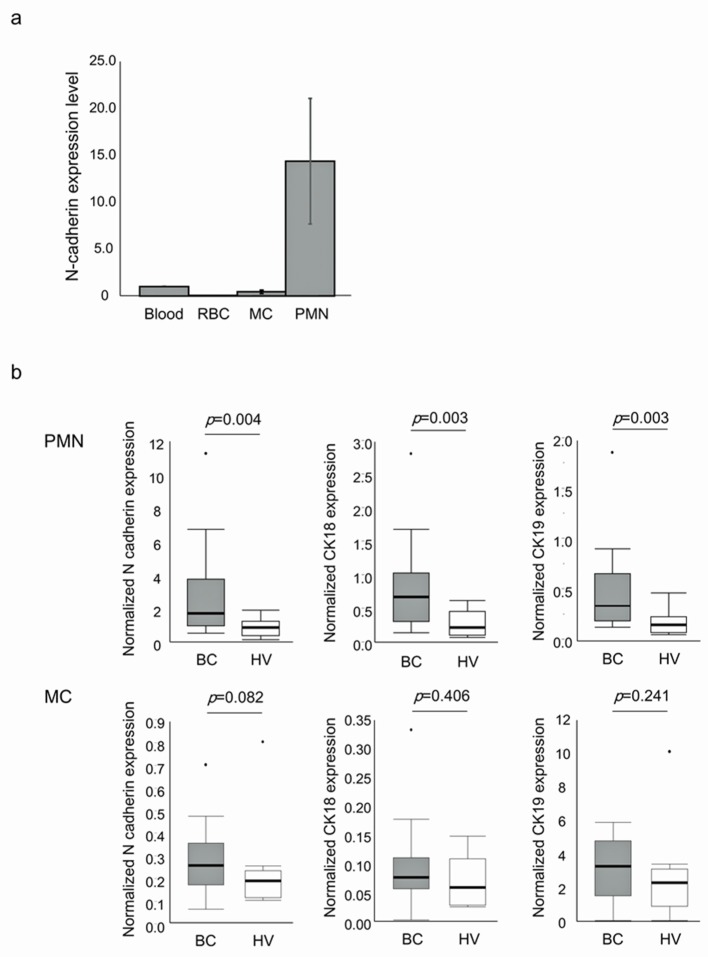
N-cadherin mRNA expression in different PB cell fractions. (**a**) N-cadherin expression in three PB cell types in breast cancer patients. RBC: red blood cells; MC: mononuclear cells; PMN: polymorphonuclear leukocytes. (**b**) N-cadherin, CK18, and CK19 expression in the PMNs and MCs of breast cancer patients and healthy volunteers.

**Table 1 ijms-21-00511-t001:** Characteristics of the 75 study patients.

**(a) Total (*n* = 75)**		
**Factor**	**Eribulin (*n* = 56)**	**S-1 (*n* = 19)**
Age, years		
Median (range)	57 (40–72)	59 (33–83)
Number of prior chemotherapy lines		
Median (range)	3 (0–8)	0 (0–3)
Luminal ^a^		
n (%)	29 (51.8)	8 (42.1)
Luminal/HER2		
n (%)	3 (5.4)	5 (26.3)
HER2-enriched ^b^		
n (%)	6 (10.7)	2 (10.5)
TN^c^		
n (%)	18 (32.1)	4 (21.1)
**(b) Patients treated with both Eribulin and S-1**		
**Factor**	**S-1 Followed by Eribulin (*n* = 35)**	**Eribulin followed by S-1 (*n* = 21)**
Age, years		
Median (range)	59 (43–72)	55 (40–71)
Number of prior chemotherapy lines		
Median (range)	3 (1–8)	0 (0–5)
Luminal ^a^		
n (%)	16 (45.7)	13 (61.9)
Luminal/HER2		
n (%)	2 (5.7)	1 (4.8)
HER2-enriched ^b^		
n (%)	6 (5.8)	0 (0.0)
TN^c^		
n (%)	11 (31.4)	7 (33.3)

^a^ Luminal, ER, or PgR-positive ^b^ HER2-enriched, only HER2-positive ^c^ TN, triple negative (ER/PgR/HER2-negative).

**Table 2 ijms-21-00511-t002:** Type of distant metastasis progression in patients undergoing S-1 or eribulin treatment.

**(a) Total (*n* = 75)**					
**Agent**	**Disease Progression**	**TTF ^a^, Median (Range)**	**NM (+) ^b^**	**NM (−) ^c^**	***p***
S-1 (*n* = 19)	16	8 (2–56)	8 (50.0%)	8 (50.0%)	0.043
Eribulin (*n* = 56)	38	6 (1–43)	7 (22.6%)	31 (77.4%)	
**(b) S-1 followed by eribulin (*n* = 35)**					
**Agent**	**Disease Progression**	**TTF, Median (Range)**	**NM (+)**	**NM (−)**	***p***
S-1	29	10 (3–59)	14 (48.3)	15 (51.7)	0.025
Eribulin	27	6 (1–22)	5 (18.5)	22 (81.5)	
**(c) Eribulin followed by S-1 (*n* = 21)**					
**Agent**	**Disease Progression**	**TTF, Median (Range)**	**NM (+)**	**NM (−)**	***p***
S-1	4	7 (1–11)	1 (25.0)	3 (75.0)	1.000
Eribulin	11	5 (1–16)	2 (18.2)	9 (81.8)	

^a^ TTF, time to treatment failure; ^b^ NM (new metastasis) (+), NM or NM + PEM (pre-existing metastasis); ^c^ NM (−), PEM only.

**Table 3 ijms-21-00511-t003:** Relationships between clinicopathological factors and N-cadherin expression in PB of breast cancer patients.

Variable	Low Expression (*n* = 167)	High Expression (*n* = 159)	*p*
	*n* (%)	*n* (%)	
Age (years)			0.092
<65	119 (71.3)	126 (79.2)	
≥65	42 (25.1)	28 (17.6)	
unknown	6 (3.6)	5 (3.2)	
Pathological tumor size			0.634
1	90 (53.9)	82 (51.6)	
2, 3	75 (44.9)	76 (47.8)	
unknown	2 (1.2)	1 (0.6)	
Nuclear grade			0.956
1, 2	108 (64.7)	105 (66.0)	
3	49 (29.3)	47 (29.6)	
unknown	10 (6.0)	7 (4.4)	
Venous involvement			0.402
(−)	155 (92.8)	143 (89.9)	
(+)	8 (4.8)	11 (6.9)	
unknown	4 (2.4)	5 (3.1)	
Lymphatic involvement			0.070
(−)	111 (66.5)	91 (57.2)	
(+)	52 (31.1)	65 (40.9)	
unknown	4 (2.4)	3 (1.9)	
Lymph node metastasis			0.194
(−)	112 (67.1)	95 (59.7)	
(+)	55 (32.9)	63 (39.6)	
unknown	0 (0.0)	1 (0.6)	
ER			0.721
(−)	41 (24.6)	42 (26.4)	
(+)	124 (74.3)	116 (73.0)	
unknown	2 (1.2)	1 (0.6)	
PgR			0.159
(−)	62 (37.1)	72 (45.3)	
(+)	103 (61.7)	87 (54.7)	
unknown	2 (1.2)	0 (0.0)	
HER2			0.235
− (0, 1)	104 (62.3)	97 (61.0)	
+ (2, 3)	44 (26.3)	55 (34.6)	
unknown	19 (11.4)	7 (4.4)	
Subtype			0.076
HR ^a^+/HER2−	92 (55.1)	76 (47.8)	
HR±/HER2+	44 (26.3)	55 (34.6)	
TN ^b^	12 (7.2)	21 (13.2)	
unknown	19 (11.4)	7 (4.4)	

Correlations were analyzed by Fisher’s exact test. ^a^ HR, hormone receptor; ^b^ TN, triple negative.

**Table 4 ijms-21-00511-t004:** Univariate and multivariate analyses of prognostic factors for recurrence-free survival (RFS) in breast cancer patients.

Variables	Univariate		Multivariate	
	HR (95% CI ^a^)	*p*	HR (95% CI)	*p*
Age (years) (≥65/<65)	0.410 (0.065–1.432)	0.181		
Pathological tumor size (2 or 3/1)	2.064 (0.845–5.494)	0.113		
Nuclear grade (3/1 or 2)	1.673 (0.648–4.135)	0.277		
Venous involvement (+/−)	1.862 × ^−9^ (0.609–1.609)	0.119		
Lymphatic involvement (+/−)	4.127 (1.655–11.653)	0.002		
Lymph node metastasis (+/−)	4.199 (1.685–11.860)	0.002	4.023 (1.610–11.380)	0.003
ER (+/−)	0.497 (0.205–1.268)	0.138		
PgR (+/−)	0.363 (0.136–0.887)	0.026	0.380 (0.142–0.931)	0.034
HER2 (2 or 3/0 or 1)	2.13 (0.875–5.210)	0.094		
N-cadherin expression in PB (high/low)	2.611 (1.046–7.380)	0.040	2.215 (0.882–6.295)	0.092

^a^ CI, confidence interval.

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
