# Peer review of "N-Cadherin mRNA Levels in Peripheral Blood Could Be a Potential Indicator of New Metastases in Breast Cancer: A Pilot Study"

_ijms, 2020, doi:10.3390/ijms21020511_

Round 1
Reviewer 1 Report
The topic is interesting. Starting from the observation that the prognosis is worse for patients affected by metastatic breast cancer that evolves through new metastasis (NM) respect to a cancer driven by a pre-existing metastasis (PEM), the authors conducted studies to discover blood biomarkers linked to NM in breast cancer. They conducted a variety of analyses of expression of epithelial and mesenchymal markers in peripheral blood of cancer patients treated with eribulin or S-1. In the end the authors find out that N-cadherin mRNA levels in blood may represent an original biomarker linked to NM in patients affected by breast cancer.
This manuscript is well written and technically sound and deserves publication in the International Journal of Molecular Science.
I suggest the following minor revisions:
page 89 "led to us" replace with "led us"
Fig.1 has rather low resolution please fix
Fig.2 The caption needs to be written more accurately. For example in panels C it's not clear what the histograms represent. Moreover, I don't like that panel d is inserted before panels C.
line 202 "using using" please fix
line 262 "five exposed ectodomain" replace ectodomain with ectodomains
lines 431-432 "...breast cancer as well as a better understanding..." I would say ..." ...treatment of breast cancer, and provide a better understanding...."
Author Response
The topic is interesting. Starting from the observation that the prognosis is worse for patients affected by metastatic breast cancer that evolves through new metastasis (NM) respect to a cancer driven by a pre-existing metastasis (PEM), the authors conducted studies to discover blood biomarkers linked to NM in breast cancer. They conducted a variety of analyses of expression of epithelial and mesenchymal markers in peripheral blood of cancer patients treated with eribulin or S-1. In the end the authors find out that N-cadherin mRNA levels in blood may represent an original biomarker linked to NM in patients affected by breast cancer.This manuscript is well written and technically sound and deserves publication in the International Journal of Molecular Science.
A: Thank you very much for your comments.
Q1: page 89 "led to us" replace with "led us"
A1: We replaced “led to us” to “led us”.(line 89)
Q2: Fig.1 has rather low resolution please fix
A2: We fixed to Fig 1 with high resolution. The resolution is 600dpi. (Figure 1)
Q3. Fig.2 The caption needs to be written more accurately. For example in panels C it's not clear what the histograms represent. Moreover, I don't like that panel d is inserted before panels C.
A3: First, we revised the caption of Fig 2 as follows; “Expression of epithelial and mesenchymal markers in the PB of breast cancer patients undergoing chemotherapy. a. A comparison of expression levels of the markers in PB between breast cancer patients just before eribulin or S-1 chemotherapy and healthy volunteers (HVs). b. Changes in the expression levels of the markers in representative cases of NM and PEM over the treatment course of eribulin or S-1 treatment. Left: case #5 treated with S-1; right: case #9 treated with eribulin. The expression levels are expressed relative to the level at pre-treatment (baseline; 1.0). BS; baseline. c. A comparison of expression levels of the markers in PB of breast cancer patients with NM or PEM. The expression levels are expressed relative to the level at pre-treatment (baseline; 1.0). BS; baseline. n; the total number of samples from patients. d. A comparison of expression levels of the markers in PB of breast cancer patients undergoing eribulin or S-1 treatment. The expression levels are expressed relative to the level at pre-treatment (baseline; 1.0). BS; baseline. n; the total number of samples from patients. e. Changes in N-cadherin expression in breast cancer patients over the course of eribulin or S-1 treatment. The expression levels are expressed relative to the level at pre-treatment (baseline; 1.0). BS; baseline” (lines 163-175)
Second, we reconstructed Fig 2 (Figure 2). Then we changed to “We compared the expression levels of epithelial and mesenchymal markers in the PB of patients with recurrent breast cancer with NM (the total number of samples; n = 9 from 4 patients) versus PEM (the total number of samples; n = 16 from 3 patients) using Ueo cohort. The changes in the expression levels of the markers in a representative case of NM or PEM are shown in Fig. 2b. Interestingly, the expression level of N-cadherin increased consistently with time in the NM case, although the level of the tumor markers CEA and CA15-3 were not elevated. In the PEM case, the expression of N-cadherin was low, although CA15-3 was elevated. Furthermore, the expression levels of the mesenchymal markers tended to be higher in the NM, whereas epithelial marker expression was higher in the PEM cases (Fig. 2c). Statistical differences in N-cadherin, vimentin, and CK18 levels between the NM and PEM cases were found (p = 0.002, p = 0.027, and p = 0.011, respectively).” (lines 136-145). Moreover, we modified the caption of Fig 2b and Fig 2c as described above. (lines 165-171)
Q4: line 202 "using using" please fix
A4: We fixed it. (line 202)
Q5: line 262 "five exposed ectodomain" replace ectodomain with ectodomains
A5: We replaced "five exposed ectodomain" with “ectodomain with ectodomains”. (lines 261-262)
Q6: lines 431-432 "...breast cancer as well as a better understanding..." I would say ..." ...treatment of breast cancer, and provide a better understanding...."
A6: We replaced “as well as” with “, and provide”. (lines 430-431)
Reviewer 2 Report
This new manuscript is a reinterpretation of the old manuscript with a new discussion as suggested by this referee. The new discussion highlights the importance of the relationships between the many post-translational modifications of the N-cadherin and the many biological activities it possesses even in pathological situations such as breast cancer. This referee expresses a favorable opinion for the publication of the manuscript.
Author Response
This new manuscript is a reinterpretation of the old manuscript with a new discussion as suggested by this referee. The new discussion highlights the importance of the relationships between the many post-translational modifications of the N-cadherin and the many biological activities it possesses even in pathological situations such as breast cancer. This referee expresses a favorable opinion for the publication of the manuscript.
A: Thank you very much for your comments.